# Modeling of injury severity of distracted driving accident using statistical and machine learning models

Neero Gumsar Sorum ⬤*, Martina Gumsar Sorum

Department of Civil Engineering, North Eastern Regional Institute of Science & Technology, Nirjuli, Arunachal Pradesh, India

* neerogsorum@gmail.com

## Abstract

Distracted Driving (DD) is one of the global causes of high mortality and fatality in road traffic accidents. The increase in the number of distracted driving accidents (DDAs) is one of the concerns among transportation communities. The present study aimed to examine the individual and interacted effects of the influential factors on the injury severity of the DDAs using the Binary Logistic Regression (BLR) method, and at the same, to select the best machine learning (ML) model in predicting the injury severity of the DDA. The selection of the best ML model was based on the optimum combination of accuracy, F1 score, and area under curve metrics. Ten years of DDA data (2011−2020) provided by the police department of Imphal, India, was used in the present study. The BLR model-without-interaction results revealed that out of twenty categorical variables, nine categorical variables (below 18, 18−24, 25−40, above 40 years age group, two-wheeler, heavy motor vehicle, 12AM-6AM, 6PM-12AM, and hit-object collision) were statistically significant to the injury severity of the DDAs. In interaction model results, there were 11, 1, and 1 significant combinations among categorical variables in two-way, three-way, and four-way interaction models, respectively. The ML model results showed that overall, the XGBoost model was reported as the best-performing model in the first hyperparameter set, and the Single Layer Perceptron model in the second set. These results may be useful for transportation policymakers while implementing any countermeasures to improve road safety in hilly areas.

## 1. Introduction

It was reported that globally 1.19 million road users (pedestrians, cyclists, passengers, and motorcyclists) lost their lives in road traffic accidents (RTAs) [1]. RTAs remain a major global issue, accounting for the 12th biggest cause of mortality as per the report of the World Health Organization [1]. Distracted driving accident (DDA)

**Data availability statement:** "All relevant data are within the paper and its Supporting Information files."

**Funding:** The author(s) received no specific funding for this work.

**Competing interests:** The authors have declared that no competing interests exist.

is one of the concerns. In recent years, the increase in the number of DDAs led to great concern among transportation policymakers and researchers. The DDAs occur mainly due to driver distraction. "*Driver distraction is a specific type of driver inattention that occurs when drivers divert attention from the driving task to focus on some other activity*" [2]. These activities may be (i) looking away from the roadway: seeing billboards (visual distraction), (ii) taking ears off the road: talking using a cell phone, listening to passengers (auditory distraction), (iii) taking hands off the steering wheel: manually adjusting the radio volume (biomechanical distraction), or (iv) taking the mind off the road: thinking (cognitive distraction) [3,4]. Mobile messaging while driving is treated as the most unsafe driver distraction as it combines cognitive, visual, and biomechanical distractions. Distracted driving (DD) endangers not just car occupants, but also pedestrians and bikers.

The DD, due to mobile phone use, billboard presence, sleepiness, and in-built vehicle systems factors, was reported to have a detrimental effect on driving performance [5]. The probability of a traffic accident occurrence was very high among distracted drivers [6,7]. It was reported that the probability of distracted drivers being involved in road accidents was four times higher than that of non-distracted drivers [8]. Recently, distracted drivers were shown to be three times as likely to cause fatal accidents than non-distracted drivers [9]. Further, the DD was found to be the 2nd leading cause of driver fault in fatal and non-fatal accidents in Jordan [10]. In 2022, 11% of all police-reported traffic accidents were recorded as DDAs in the United States [2]. Moreover, the National Safety Council reported that DDAs were found to be equivalent to road accidents that occurred due to alcohol consumption and over-speeding factors in terms of fatal and serious injury levels [11]. Therefore, the current study focuses on the evaluation of the influential factors and prediction of accident severity due to DD.

The issue of road safety became even more important for India because the country shares 11% of global RTAs, as per the report of the Ministry of Road Transport and Highway [12]. There were in total 0.15 million deaths and 0.38 million injuries reported from 0.41 million RTAs in India. Further, it was reported that 9.5% of total accidents in India were due to drunken driving, jumping at red light, and use of mobile phones, resulting in 9.8% of total deaths [12]. The current study was conducted in Imphal city of Manipur, a hilly state in the North-East region of India. The state has 3 million people with 128 per sq km as population density. The city is located at 24.8°N, 93.94° E, and has a height of 786 meters from the mean sea level. Four-lane divided National Highway-37 passes through Imphal city. Manipur state stands in 25th position (with accident severity: 30) for contributing the highest number of road accident deaths per 100 accidents [12]. Therefore, the current study has set two objectives: (i) to examine the individual and interacted effects of the contributing factors on the injury severity of the DDA that occurred in Imphal city using a logistic regression method, and (ii) to select the best machine learning (ML) model in predicting the injury severity of the DDA.

The next part (section 2) of the manuscript provides a review of pertinent literature. Section 3 illustrates the detailed explanation of the study data and methods.

Modeling results and discussion are presented in section 4. Section 5 concludes with a summary, findings, and recommendations for further study.

## 2. Literature review

In the field of DD, most of the researchers have focused their studies on (i) identifying the underlying causes/sources of DD [13–16]; (ii) the impact of DD behaviors on driving performance [17–21], drivers' physiological and visual signals [22–25], and traffic flow [26,27]; (iii) its association with accidents or accident risk [8,28–30]; and (iv) driver distraction detection using ML and Deep Learning (DL) [31–34]. The summary of some DD studies is tabulated in Table 1. The following subsections outline the previous study results of DD where either the conventional statistical or ML method was used.

### 2.1. Studies based on distracted driving using statistical models

In the United States, a multinomial logit model was developed to forecast cognitive, in-built vehicle, passenger, and cellphone-related distractions [30]. The authors reported that teenage drivers were more prone to rear-end and angular accidents due to cognitive attention than stationary-object collisions.

Neyens & Boyle [35] developed an ordered logit model to forecast the distracted drivers' and their passengers' injury severity in the United States. They found that teenage drivers had a higher probability of being severely injured if they were distracted by a mobile phone or by passengers than if the distraction was caused by in-vehicle technology or by the driver's inattention.

Cooper et al. [26] reported that the drivers, distracted due to mobile phone use, had less tendency to change lanes and drove at slower speeds and thus spent more time tailgating, regardless of traffic flow. They reported that DD had an adverse effect on traffic flow. Stavrinos et al. [27] extended Cooper's work by incorporating drivers of the young age group. The results of the study were quite consistent with those reported by [26].

Hosking et al. [13] evaluated the impacts of cellphone usage (retrieving and texting) on the driving performance of young novice drivers using a mixed-model two-way Analysis of Variance (ANOVA) in Australia. The authors found that texting while driving increased driver's variability in lane position by 50%. There was also an increase in headway value by 150%.

Liu & Donmez [36] built an ordered logit model for the examination of the relation between injury severity and driver distraction in Canada. They found that cognitive distractions decreased driver injury severity, whereas in-vehicle distractions increased the injury severity of distracted drivers.

Zhu & Srinivasan [37] evaluated the association between the influential factors and injury severity of the DDA using an ordered-probit model in the United States. They observed that distracted truck drivers were found to be highly associated with higher accident severities.

Hanley & Sikka [38] used a multinomial logit model to examine the relationship between driver injury severity and driver distraction in the United States. They observed the injury severity of distracted drivers was less compared to that of non-distracted drivers.

In Australia, Beanland et al. [14] reported that the most common sources of distraction were voluntary and originated from inside the vehicle. They noted that 20% of the DDAs were due to passenger interaction distractions. This was consistent with the result of a UK study conducted by [39]. The authors reported that passenger interactions were the most common distraction in severe accidents.

Donmez & Liu [40] built an ordered logit model to predict driver injury severities of the DDAs in Canada. Results revealed that the injury severity of young and middle-aged drivers increased when they were involved in cellphone dialing/texting, using in-built vehicle systems, and interacting with passengers while driving.

Haque & Washington [19] compared the breaking behavior of mobile-distracted and non-distracted drivers using the Weibull regression model in Australia. They found that distracted drivers were found to apply vehicle brakes abruptly and excessively compared to non-distracted drivers, posing major safety issues for vehicles behind them in a traffic stream.

**Table 1. Summary of some distracted driving studies.**

| Authors Name | Parameters/ metrics | Method/Algorithm | Key finding |
|---|---|---|---|
| Torkkola et al. [64], USA | Accuracy | RF and Quadratic classifier | The RF classifier outperformed the quadratic classifier |
| Neyens & Boyle [35], USA | Injury severity | Ordered logit model | Passengers and adolescent drivers were more likely to be injured when their driver was distracted by electronics or passengers than when the driver was neither distracted nor inattentive. |
| Sathyanarayana et al. [65], USA | Accuracy | KNN | The KNN model was able to detect driver attention with a 90% accuracy rate. |
| Al-darrab et al. [17], Saudi Arabia | drivers' reaction time in braking | ANOVA | The duration of a cellphone call had the greatest impact on brake reaction time. |
| Miyaji et al. [66], Japan | Accuracy | SVM and AdaBoost | The AdaBoost model (93%) outperformed the SVM model (92%) |
| Yannis et al. [21], USA | Speed and headway | Linear and log-linear regression methods | Driving speed reduced, and headway increased while using a mobile phone |
| Liu & Donmez [36], Canada | Injury severity level | Ordered logit model | The injury severity of police drivers in DDA was found to be more than that of civilian drivers. |
| Owens et al. [20], USA | Reaction time, glance | General Linear Model | Older drivers spend more time texting and looking inside. |
| Donmez & Liu [40], Canada | Injury severity level | Ordered logit model | Injury severity of older drivers was found to be the highest when they were involved in mobile dialing or texting. |
| Xian & Jin [67], China | Eye fixation, vehicle control, completion time | ANOVA | When performing e-mail duties, drivers spent 2.41 times more time gazing away from the road. |
| Thapa et al. [68], USA | Speed, lane deviation | F-test | Significant lateral lane deviations remained for an average of 3.38 seconds after texting was ceased. |
| Dingus et al. [29], USA | Accident risk | Mixed effect random logistics method | The likelihood of an accident was enhanced while dealing with built-in car systems. |
| Choudhary & Velaga [15], India | Reaction time | Weibull regression method | Mobile talk and texting increased drivers' reaction time in both pedestrian and road crossing events |
| Gao & Davis [69], USA | Reaction time | ANOVA | Driver distraction duration increased reaction time. |
| Masood et al. [70], India | Accuracy | CNN | The CNN model could easily detect driver distraction with an accuracy value of 99%. |
| Mollu et al. [71], Belgium | Glance behavior, speed, reaction time | ANOVA | More eye contact during brief billboard message display times. The presence of a digital billboard increased the braking reaction time by 1.5 times. |
| Jeong & Liu [72], USA | Lane-keeping, eye movement | ANOVA | Mobile distraction causes drivers to be unable to maintain a consistent lateral lane position on curve roads. |
| Meuleners et al. [73], Australia | Speed, lane deviation, headway, gaze fixation | ANOVA | The presence of digital billboards impaired driving performance (driving speed, headway, lane position, and visual fixations) |
| Garcia-Herrero et al. [74], Spain | Injury severity | BN | Technology-based distractions increased the probability of driver infractions, leading to serious accidents |
| Puspasari et al. [75], Indonesia | Violations, errors, and lapses | Partial Least Squares Structural Equation Model and Pearson's Chi-square test | Male drivers were more distracted by attractive roadside objects compared to females. |

The distracting activities of teen drivers in rear-end DDAs were examined in the United States using chi-square and Student's t-tests [41]. The authors observed that 50% and 10% of the rear-end DDAs were contributed by frequent passenger interactions and cell phone use activities. It was also reported that distracted drivers took a longer response time to apply brakes as compared to non-distracted drivers.

Choudhary & Velaga [18] studied the impacts of cell phone conversation and messaging on the driving performance of inexperienced young drivers and professional drivers using multiple linear regression models in India. The authors found that the risky driving behavior of distracted drivers deteriorated their driving performance (longitudinal and lateral control of the vehicle). The degradation rate was higher in the case of inexperienced young drivers.

The study by Atwood et al. [42], conducted in the United States, confirmed that drivers who were involved in mobile texting at a greater rate were more likely to be involved in an accident. The results of negative binomial regression models revealed that the rate of severe accidents increased by 0.58% for every additional text per day and 8.3% per text per hour of driving. They also found that young drivers texted significantly more than middle-aged and older drivers.

Qin et al. [43] investigated the impacts of various distraction sources on fatal accidents in the context of the age and gender of drivers in the United States using Tukey and chi-square test. They found that cognitive distraction dominated a large proportion of driver secondary tasks. Further, it was also reported that young female drivers had a higher likelihood of being distracted by in-built vehicle systems compared to males.

Choudhary et al. [28] conducted a comparative analysis of the effect of overall driving performance on accident risk using a structural equation model in India. They observed that the drivers who were involved in mobile texting while driving were more likely to have a serious accident than drivers who were involved in visual-manual tasks related to music players.

Lu et al. [44] evaluated the impacts of mobile distraction on RTAs using the propensity score method in the United States. The authors observed that cellphone talking increased the accident risk for young drivers. Further, it was reported that cellphone distraction due to visual-manual tasks had a higher probability of accident occurrence when compared to overall cellphone distraction and cellphone talking.

G. Zhang et al. [45] examined the impacts of DD on at-fault and innocent drivers' injury severities using the propensity score method in China. They found that drivers were more likely to be distracted in scenarios including higher speed limits, gloomy weather, intersections, young and female drivers, heavier vehicles, and non-peak hours. The injury severities of the DDAs for at-fault drivers were found to be higher than that of the innocent drivers.

Arevalo-tamara et al. [16] applied a structural equation model to examine the various risky driver behaviors that lead to DD, and, in turn, severe RTAs in Bogota, Columbia. They found that the main source of DD among young drivers (below 25 years old) was cell phone use while driving. These drivers had a higher tendency to commit more traffic violations, leading to severe traffic accidents.

## 2.2. Studies based on distracted driving using ML models

Several ML algorithms, namely Random Forest (RF), Decision Tree (DT), Logistic Regression (LR), Bayesian Network (BN), eXtreme Gradient Boosting (XGBoost), Convolutional Neural Network (CNN), Single Layer Perceptron (SLP), Support Vector Machine (SVM), Ada-Boost, Naive Bayes (NB), Artificial Neural Network (ANN), and K-Nearest Neighbors (KNN), have been employed in various distraction studies. Liang et al. [46] compared the performance of the SVM and LR models in driver distraction detection based on accuracy metric in the United States. Results revealed that the SVM model was found to be a more accurate model for the detection with an accuracy value of 81.1% when compared to the LR model (accuracy = 72.7%). A similar finding was reported in studies by Tango & Botta [47] and Son et al. [48]. Tango & Botta [47] reported that the SVM model outperformed the other ML (Neural Network) model in the detection system of driver distraction. Son et al. [48] observed that the proposed SVM models showed excellent performance in the identification of driver's cognitive distraction (accuracy = 89.0%, sensitivity = 86.4%, and specificity = 91.7%). In another study, Liao et al. [49] examined the efficacy of SVM models in identifying driver cognitive distraction at stop-controlled crossings and speed-limited highway sections. They observed the accuracy obtained from the SVM model for the stop-controlled intersection (95.8%) was higher than that of the speed-limited highway section (93.7%).

In Canada, Ragab et al. [50] used several ML algorithms (Adaboost, hidden Markov, RF, SVM, conditional random field, and ANN) for visual-based driver distraction detection. In comparison, the RF algorithm showed a superior performance in distraction detection.

One of the objectives of Wiener's [51] work was to differentiate and classify cognitive distraction, sensorimotor distraction, and normal driving of drivers using the BN and RF algorithms. The author observed that both the algorithms were able to differentiate and classify accurately the various targeted distractions. In terms of performance comparison, the RF model outperformed the BN model. Further, it was found that driving performance was more informative than driver physiology in the context of classifying driver distraction.

Koohestani et al. [22] implemented four classification techniques (RF, DT, Generalised Linear Model Net Classification Algorithm, and Recursive Partitioning Classification Algorithm) to forecast lane deviation variation using biological and physiological data collected from Australian drivers. They found that the RF classifier achieved the best performance among the other classifiers. In Brazil, Torres et al. [31] developed seven ML models (SVM, DT, LR, RF, AdaBoost, Gradient Boosting Machines (GBM), and CNN) to automatically differentiate between drivers and passengers while reading a mobile text in a vehicle. They found that the CNN and GBM models were found to be the most effective in distinguishing passengers and drivers in the event of reading while driving.

Gjoreski et al. [32] compared the performance of seven ML algorithms (RF, DT, SVM, KNN, XGBoost, AdaBoost, and Bagging) and seven DL algorithms to detect driver distraction. They concluded that the XGBoost model achieved the best performance among the compared models. In a study, Ma et al. [52] applied the SVM model to examine driver distractions in terms of driving performance measures (horizontal and vertical control) in Harbin city, China. Results revealed that the SVM model evaluated effectively the degree of the driver's visual and cognitive distraction with an accuracy value of 89.9%. In another study, Mcdonald et al. [53] trained seven ML algorithms to find out the best ML model for driver distraction detection and prediction of the distraction source. They found that the RF was observed to be the highest-performing ML model for accurately classifying driver distraction. Sun et al. [54] found that the proposed model, bidirectional long short-term memory network (Bi-LSTM) combined with an attention mechanism (Atten-BiLSTM), outperformed the SVM model and the long short-term memory network models, with the highest recognition accuracy of 90.64% for the prediction of cognitive distraction.

Ahangari et al. [55] used a BN model to detect driver distraction in terms of driving performance measures in a driving simulator. They reported that the BN model presented the best performance for the driver distraction prediction problem based on the accuracy (67.8%), sensitivity (62.6%), and area under curve (AUC) (75.1%) metrics. Al-Doori et al. [56] used an open-access image dataset to detect driver distraction using three ML models: KNN, SVM, and RF models. They found that the KNN model showed the best performance with an accuracy value of 98.1%. This was followed by the SVM model (95.8%) and the RF model (88.7%). Bano et al. [57] carried out a comparative analysis of drowsy driver detection using three ML algorithms: SVM, CNN, and Hidden Markov Model (HMM). The HMM model showed a better performance compared to the SVM and CNN models. Chai et al. [58] studied driving distraction detection using the XGBoost model in terms of lane-keeping performance. The results showed that the XGBoost model achieved satisfactory results in driving distraction detection, with a predictive accuracy of 90% at the individual level. Z. Zhang et al. [59] used MATLAB software to implement four ML algorithms (DT, SVM, KNN, and LR) for driver distraction detection in the United Kingdom. The authors found the DT model as the better model to detect driver distraction when compared to the SVM, KNN, and LR in terms of accuracy and computation time. The DT model showed a 78.4% accuracy capacity of giving timely warnings to the driver in real-time.

Aljasim & Kashef [60] explored the performance of ResNet50, VGG16, MobileNet, and Inception models to detect distracted driver actions; and proposed a new model, E2DR, to improve accuracy, enhance generalization, and reduce overfitting. The authors that the ResNet50 model outperformed the VGG16, Inception, and MobileNet models with an accuracy value of 88%. Further, the highest-performing E2DR model was the ResNet50-VGG16 variant with a 92% accuracy value. Bachtiar et al. [61] conducted a driver distraction detection study by comparing the performance of RF, KNN, SVM, and CNN models in terms of accuracy and training speed. The RF model obtained the best performance with an accuracy value of 98.85%, and the CNN

model showed the lowest performance (accuracy = 97.23%). In terms of time to recognize the distraction activity, the KNN demonstrated the fastest model to recognize the activity distraction, while the SVM was the least one. Misra et al. [33] studied driver cognitive distraction detection by implementing six ML algorithms (RF, SVM-linear, SVM-RBF, KNN, NB, and DT). The study results illustrated that the RF model presented the best performance among all other trained models in terms of accuracy metric (average accuracy value > 90%). The NB model showed the worst performance in distraction detection.

In a study by Ali & Haque [34], the variation in response times of mobile-distracted young drivers while driving was investigated using a hybrid modeling framework (DT and random parameters duration model). The modeling results showed that mobile phone distraction impaired response time behavior for the majority of drivers. However, some drivers tend to respond earlier while being distracted, especially the female group. In another study, Javid et al. [62] used the BN model to examine the impact of mobile phone blocking applications on driving behaviors and accidents caused by DD in the United States. The authors found that if all drivers used mobile phone blocking applications, mobile DDAs would be reduced by 5%, and self-reported distraction by 9%. Sun et al. [63] used an association rule mining technique to evaluate the contributing factors to the DD. They observed that the nighttime (18:00–23:59), speed (70–80 kmph), travel time (1–3 hours), freeways, acceleration (< 0.5 m/s$^2$), visibility (> 1 km), and tangent roadway section were highly associated with the DD.

In summary, many researchers focused their studies on the impact of DD to assess accident risk and driving performance measures, and driver distraction detection using ML and deep learning techniques. Most of these studies were conducted in the USA and other developed countries. Moreover, the identification of the contributing factors and model prediction of injury severity in DDAs has not been yet fully researched, particularly in India. To fill this gap, the present study aims to examine the individual and interacted effects of the contributing factors on the injury severity of the DDA using a logistic regression method, and at the same time to find out the best ML model in predicting the injury severity of the DDA using coding-free software.

## 3. Data and methods

### 3.1. Data description

Police reported single- and two-vehicle-involved DDA data, collected during 2011–2020 from Imphal city (Manipur) of the northeastern states of India, was used in the present study. All missing data and data for passengers and more than two vehicle-involved accidents were excluded from the datasets. In the present study, the DDAs were defined as those accidents which occurred due to driver engagement in secondary tasks (mobile use, chatting and talking, or looking away from the roadway) other than driving. A total of 949 observations (individual road users- either driver or pedestrian involved in DDA), after excluding all missing data, passenger data, and data for more than two vehicle-involved accidents, were available for modeling and analysis. The dataset contained two DDA categories: Non-fatal (not resulting in death within one month of the accident) and fatal (resulting in death within one month of the accident).

### 3.2. Study variables

Five independent variables, namely age, gender, vehicle type, time of accident, and nature of accident, were established in the present study. Established twenty categorical variables included four age groups (below 18, 18−24, 25−40, and above 40-year age group), two gender (Male and Female), three vehicle type (Two-wheeler, Light Motor vehicle, and Heavy Motor Vehicle), four time of accident (12AM-6AM, 6AM-12PM, 12PM-6PM, and, 6PM-12AM), and seven nature of accidents (head-on collision, side-collision, rear collision, hit object, hit pedestrian, overturn, and other). Table 2 displays descriptive statistics and notation used for established categorical variables.

### 3.3. Binary logistic regression (BLR)

The binary logistic regression (BLR) model was applied in the present study because the target variable was binary: fatal and non-fatal categories. In the present study, the objective of using this model is to identify the contributing factors of the

injury severity in the DDA, and also, to examine the interaction effect of the identified factors on the injury severity in the DDA. The mathematical expression (1) of the BLR model [76] is shown as

$$Logit(Y) = \ln\left(\frac{Y}{1-Y}\right) = \beta + b_1 x_1 + b_2 x_2 + \ldots\ldots\ldots\ldots + b_n x_n$$

(1)

where, $Y$ is the probability of occurring fatal accident, $\beta$ is constant (intercept), $b_n$ is the regression coefficient and $x_n$ is the independent variable.

For modeling, one of the categorical variables from each independent variable was deleted from the model and utilized as a reference category (as shown in Table 2). Two sets of the BLR models (Model-without-interaction and Model-with-interaction: two-way interaction and three-way interaction) were generated using Statistical Package for the Social Sciences (SPSS) software. To present the BLR Model-with-interaction approach briefly, some notations are defined (Table 2 can be referred to). The complete procedure steps of the BLR modelling are presented in the flow-chart, given in Fig 1.

### 3.4. Machine learning (ML) algorithms

Machine learning (ML) was defined as "*a field of study that gives computers the capability to learn without being explicitly programmed.*" by Arthur Samuel [77]. Supervised learning, unsupervised learning, and reinforcement learning are the three main categories of ML techniques [78]. Since the target variable (injury severity of the DDA) had a

**Table 2. Descriptive statistics and notation used for categorical variables.**

| Variables | Category | Total (%) | Injury Level (%) | |
|---|---|---|---|---|
| | | | Non-Fatal | Fatal |
| Age | Below 18[a] | 6.64 | 65.08 | 34.92 |
| | 18-24 ($A_1$) | 19.81 | 78.19 | 21.81 |
| | 25-40 ($A_2$) | 58.38 | 78.34 | 21.66 |
| | Above 40 ($A_3$) | 15.17 | 86.11 | 13.89 |
| Gender | Male[a] | 87.67 | 79.09 | 20.91 |
| | Female ($G_1$) | 12.33 | 75.21 | 24.79 |
| Vehicle Type | Two-wheeler[a] | 41.73 | 76.26 | 23.74 |
| | LMV ($VT_1$) | 50.90 | 78.47 | 21.53 |
| | HMV ($VT_2$) | 7.38 | 92.86 | 7.14 |
| Time of Accident | 12AM-6AM[a] | 4.32 | 90.24 | 9.76 |
| | 6AM3.-12PM ($T_1$) | 19.39 | 80.43 | 19.57 |
| | 12PM-6PM ($T_2$) | 40.04 | 80.00 | 20.00 |
| | 6PM-12AM ($T_3$) | 36.25 | 74.71 | 25.29 |
| Nature of Accident | Head on collision[a] | 27.08 | 76.26 | 23.74 |
| | Side collision ($N_1$) | 19.92 | 77.25 | 22.75 |
| | Rear end collision ($N_2$) | 17.49 | 79.52 | 20.48 |
| | Hit object ($N_3$) | 15.91 | 82.78 | 17.22 |
| | Hit pedestrian ($N_4$) | 12.64 | 78.33 | 21.67 |
| | Overturn ($N_5$) | 4.21 | 77.50 | 22.50 |
| | Other ($N_6$) | 2.74 | 84.62 | 15.38 |

Note:

[a]Reference category.

binary nature (fatal, and non-fatal), the classification approach was the most appropriate data mining functionality in the present study. The RF, XGBoost, DT, KNN, SVM, and SLP were the six supervised ML algorithms used in the present study (readers are requested to refer following articles: Naser [79] and Sorum & Pal [80], for the description of these algorithms). All ML algorithms were used in their default settings as available in the Dataiku platform.

### 3.5. Data pre-processing and transformation

The same dataset (949 observations) of Imphal city was used for building the ML models. The dataset was imported from an Excel file in the Dataiku platform. In the data pre-processing stage for the ML model, removal of less significant factors (e.g., year of accident and place of accident variables) were removed from the imported dataset. The entire variables in the datasets were transformed into integer values.

### 3.6. Stratified k-fold cross-validation (FCV) and train ratio (TR)

Stratified k-FCV, a most effective technique especially when dealing with unbalanced datasets, was adopted in the present study. According to this technique, the whole dataset was shuffled to ensure that the sequence of the inputs and outputs was fully random and that the inputs were not biassed in any manner. After that, the dataset was divided into k folds of equal size. The model was trained k times, with k-1 folds used for training and the rest for validation. Train Ratio (TR) term refers to the percentage of the data that is allocated to the training set when the data is divided into a training set and a test set. For example, TR value of 0.70 implies that 70% of the data is used for training an ML model, and the rest 30% of the data is reserved for testing the performance of that model. For the First set, all six ML models were implemented using 5, 10, and 15-FCVs in a TR value of 0.70 in the present study. However, in the second set, the ML models were trained at TR values of 0.70, 0.80, and 0.90 with 10-FCV.

### 3.7. Performance metrics

After the establishment of a set of independent variables from the collected traffic accident data, descriptive statistics of all the established independent variables were analyzed. The included variables for descriptive statistics analysis were driver age, gender, vehicle type, time of accident, cause of accident, nature of accident, and year of accident. Table 3 presents the percentages of fatal and non-fatal accidents of all the categorical variables. Analysis of fatal and non-fatal accident results of the categorical variables are briefly illustrated in the following subsections.

Three performance metrics, namely accuracy, F1 score, and AUC, were used to compare the predictive performances of the models used in the current study. Accuracy is the proportion of correctly classified instances out of all instances. This metric can be calculated with the following equation:

$$Accuracy = \frac{TP + TN}{TP + TN + FN + FP}$$

(2)

where, TP = True Positive, TN = True Negative, FN = False Negative, and FP = False Positive.

F1 score combines both precision and recall into a single metric, providing a balance between these two measures. The F1 score is calculated as the harmonic mean of precision and recall (equation 3).

$$F1\ Score = \frac{2 * Precision * Recall}{Precision + Recall}$$

(3)

where, precision evaluates the ML model's ability to correctly predict positive instances among all instances predicted as positive. Its value can be computed using equation 4. The recall metric evaluates the ML model's ability to correctly identify all positive instances out of all actual positive instances. This metric can be computed with the equation 5.

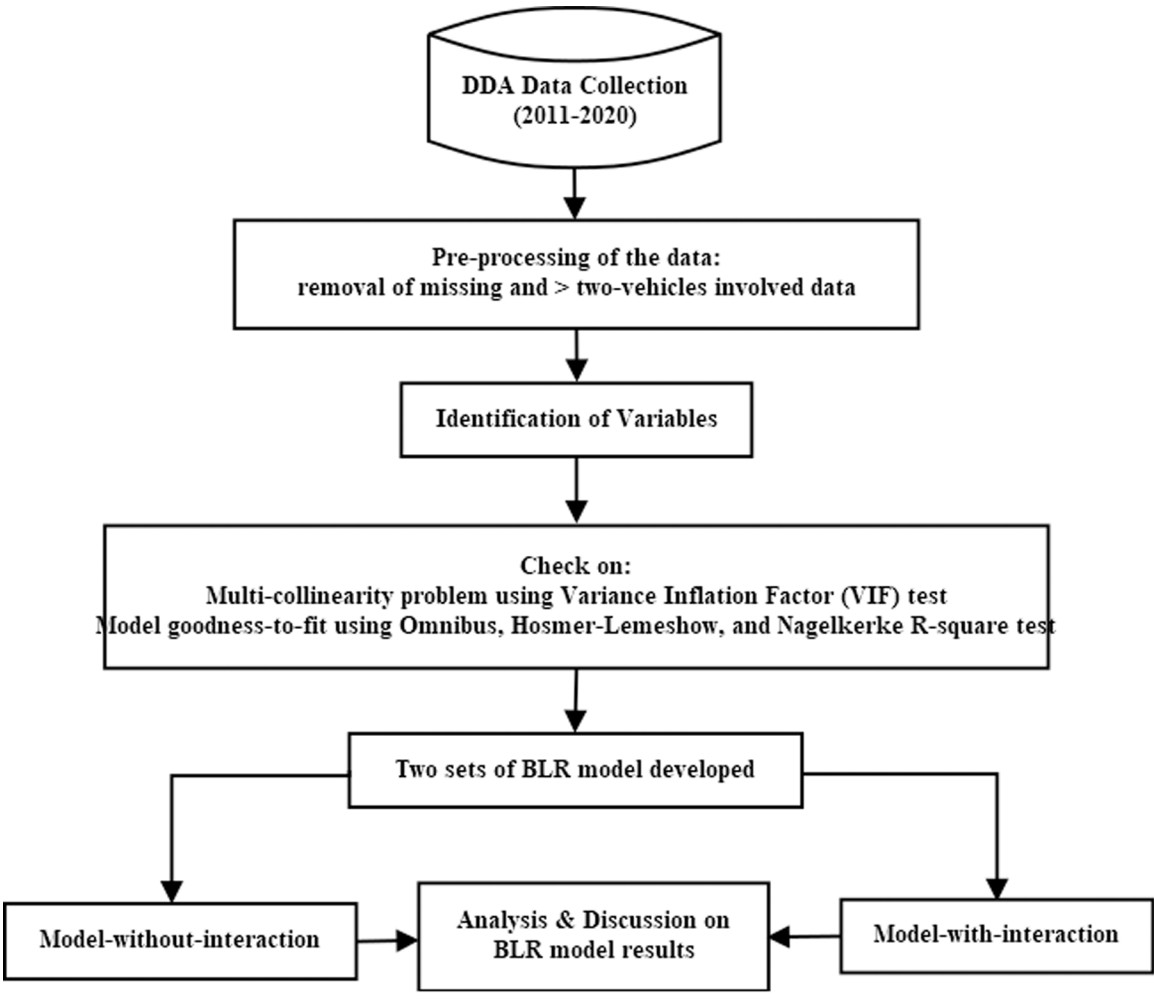

**Fig 1. The flowchart showing the procedure steps of the BLR modeling.**

**Table 3. Results of the multicollinearity test.**

Coefficients[a]

| Model | Unstandardized Coefficients | | Standardized Coefficients | t** | Sig. | Collinearity statistics | |
|---|---|---|---|---|---|---|---|
| | b | S.E. | β | | | Tolerance | VIF |
| (Constant) | 0.298 | 0.094 | | 3.166 | 0.002 | | |
| Time of accident | 0.038 | 0.016 | 0.078 | 2.422 | 0.016 | 0.998 | 1.002 |
| Nature of accident | −0.010 | 0.008 | −0.041 | −1.267 | 0.205 | 0.997 | 1.003 |
| Vehicle Type | −0.048 | 0.022 | −0.072 | −2.227 | 0.026 | 0.989 | 1.011 |
| Age | −0.048 | 0.017 | −0.089 | −2.733 | 0.006 | 0.987 | 1.014 |
| Gender | 0.038 | 0.040 | 0.031 | 0.953 | 0.341 | 0.996 | 1.004 |

[a]Dependent Variable: Injury Severity of DDA.

**t: statistics.

$$Precision = \frac{TP}{TP + FP} \qquad (4)$$

$$Recall = \frac{TP}{TP + FN} \qquad (5)$$

The area under the Receiver Operating Characteristic (ROC) curve (AUC) is the region surrounded by the ROC curve with a maximum value of one, signifying a model with perfect discrimination. The model's ability to differentiate between positive and negative situations improves as the AUC increases [81].

**3.8. Selection of the best ML model for the prediction of injury severity of the DDA**

The selection of the best ML model out of six supervised ML models was done based on the optimal combination of the three above-mentioned performance metrics (accuracy, F1 score, and AUC). The flowchart showing the procedure steps for the selection of the best ML model in the Dataiku platform is presented in Fig 2. (Readers are requested to refer to Sorum & Pal [80], for a detailed description of the research steps).

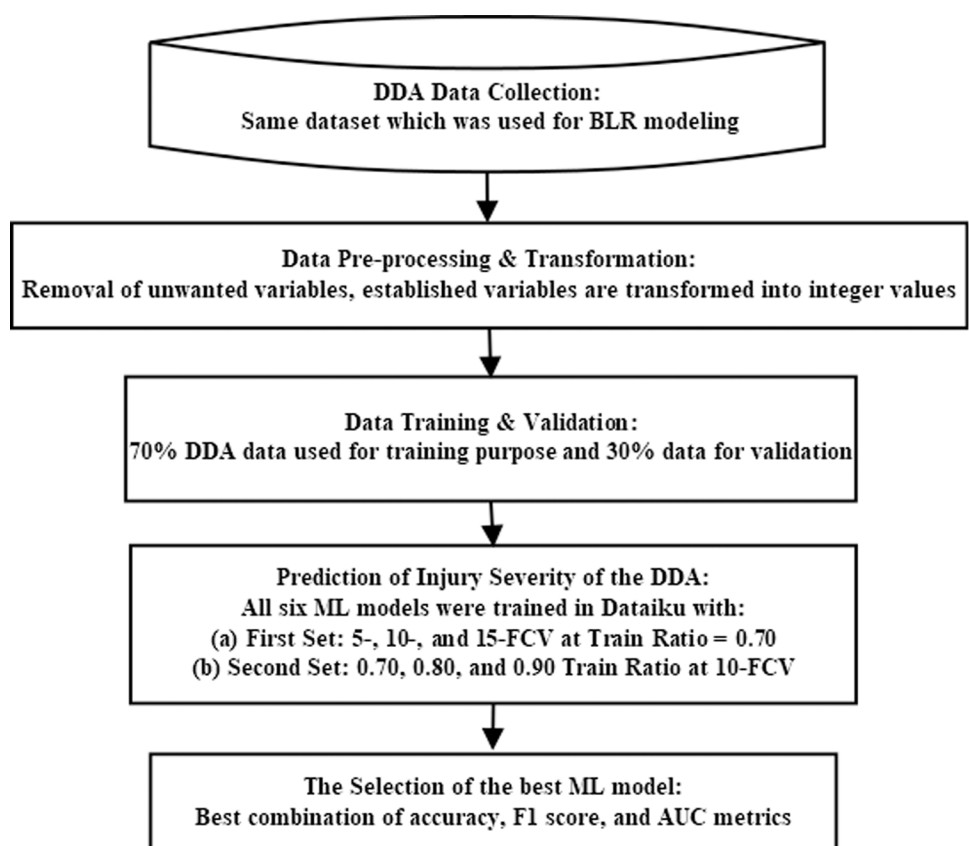

**Fig 2. Flowchart for selecting the best ML model for the prediction of DDA severity.**

# 4. Results and discussion

## 4.1. Descriptive statistics of categorical variables

In the present study, the established variables for descriptive statistics analysis were age, gender, vehicle type, time of accident, and nature of accident. The percentages of fatal and non-fatal DDAs of all the categorical variables are presented in Table 3. Overall, the road users, aged 25−40 years old, contributed the highest DDAs in the Imphal city region. However, in terms of non-fatal DDAs, the above-40 years age group road users showed the maximum percentage value (87.05%). The highest percentage of fatal DDAs was associated with the below-18 years age group road users (33.33%). The majority of road users who got into the DDAs were men (89.80%). However, females were involved more in fatal DDAs (24.47%) as compared to males (20.29%). Further, it was observed that the maximum DDAs were associated with the LMVs (50.87%), followed by two-wheelers (41.76%) and HMVs (7.38%). However, most of the HMVs were involved in non-fatal DDAs (94.12%) and two-wheelers were in the fatal DDAs (23.90%). Furthermore, the afternoon driving time (12PM-6PM) contributed the maximum percentage of the DDAs (39.15%). This was followed by night driving time (6PM-12AM) (36.98%), morning driving time (6AM-12PM) (19.52%), and dawn driving time (12AM-6AM) (4.34%). In the vehicle collision variable, the head-on collision categorical variable was found to be the most frequent collision type in the DDAs (28.09%). This was followed by side collision (20.50%), rear-end collision (17.90%), hit object (16.38%), hit pedestrian (9.87%), overturn (4.45%), and other types of collision (2.82%).

## 4.2. Multicollinearity test and BLR model fitness result

Multicollinearity exists when two or more independent variables in a model have a strong relation with one another, making it difficult to discern each independent variable's influence on the dependent variable. In the present study, variance inflation factor (VIF) values were computed to check this correlation. The results, from Table 3, revealed that the values of VIF of all the variables were less than 5. This suggested that the multicollinearity problem did not exist in the dataset.

A check on the model goodness-of-fit to the dataset was carried out in the SPSS software to assess how well the BLR model fits the data it was built upon. The Omnibus, Hosmer–Lemeshow, and Nagelkerke R-squared tests were used for this checking in the present study. To make the model statistically significant, the p-value should be less than 0.05 in the Omnibus test and more than 0.05 in the Homer-Lemeshow test [76]. A higher Nagelkerke R-squared value (from 0 to 1) shows a better fit of the model to the data [82]. The results of Table 4 illustrated that all the parameters met the minimum requirements for the model to be statistically significant. This indicated that the BLR model performed well for the dataset.

## 4.3. BLR model-without-interaction results

Table 5 illustrates the results of the BLR model-without-interaction for Imphal city. The odd ratio (OR) of the reference categorical variable is equal to 1. The eight categorical variables were statistically significant to the injury severity of the DDA at a 95% confidence level, and one categorical variable was significant at a 90% confidence level.

The results in Table 5 reveal that the age variable was observed to be statistically significant to the DDA severity at a 95% confidence level. The results also reveal that the injury severity of the DDA for the below 18-year age group road users was more than that of the 18–24, 25–40, and Above 40-years age group road users (OR =0.483, 0.490, and 0.304, respectively). One possible reason for this result could be the driver's inexperience, and their fast and aggressive driving behavior [83]. This agrees with the findings of prior investigations [40,43,45,84]. Qin et al. [43] reported that adolescent drivers were more likely to be distracted by internal sources of distraction.

**Table 4. Model goodness-to-fit.**

| Study area | Omnibus test | Hosmer–Lemeshow Test | Nagelkerke R Square |
|---|---|---|---|
| Imphal city | 0.002 | 0.206 | 0.057 |

**Table 5. Results of BLR model-without-interaction.**

| Variable | Category | b | S.E.[1] | Wald[2] | p-value | Sig* | Exp(b) |
|---|---|---|---|---|---|---|---|
| Age | Below 18 | **0** | | 10.585 | 0.014 | 95% | **1** |
| | 18-24 | −0.727 | 0.330 | 4.869 | 0.027 | 95% | 0.483 |
| | 25-40 | −0.713 | 0.294 | 5.893 | 0.015 | 95% | 0.490 |
| | Above 40 | −1.192 | 0.368 | 10.505 | 0.001 | 95% | 0.304 |
| Gender | Male | **0** | | | | | 1 |
| | Female | 0.171 | 0.242 | .500 | 0.479 | Not | 1.187 |
| Vehicle Type | Two-wheeler | | | 8.172 | 0.017 | 95% | **1** |
| | LMV | −0.060 | 0.167 | 0.129 | 0.719 | Not | 0.942 |
| | HMV | −1.379 | 0.483 | 8.151 | 0.004 | 95% | 0.252 |
| Time of accident | 12AM-6AM | 0 | | 7.525 | 0.057 | 90% | **1** |
| | 6AM-12PM | 0.854 | 0.569 | 2.253 | 0.133 | Not | 2.348 |
| | 12PM-6PM | 0.873 | 0.551 | 2.508 | 0.113 | Not | 2.394 |
| | 6PM-12AM | 1.206 | 0.552 | 4.779 | 0.029 | 95% | 3.341 |
| Nature of accident | Head on collision | **0** | | 4.654 | 0.589 | Not | **1** |
| | Side collision | −0.145 | 0.234 | .385 | 0.535 | Not | 0.865 |
| | Rear end collision | −0.240 | 0.249 | 0.933 | 0.334 | Not | 0.787 |
| | Hit object | −0.531 | 0.269 | 3.914 | 0.048 | 95% | 0.588 |
| | Hit pedestrian | −0.155 | 0.277 | 0.312 | 0.576 | Not | 0.857 |
| | Overturn | −0.091 | 0.415 | 0.048 | 0.827 | Not | 0.913 |
| | Other | −0.583 | 0.570 | 1.047 | 0.306 | Not | 0.558 |
| | Constant | −1.286 | 0.607 | 4.492 | 0.034 | | 0.276 |

a = Reference category, b = Regression Coefficient, S.E. = Standard Error, Wald = chi-squared test that tests the null hypothesis of the constant equals 0, p-value = Statistical Significance, Sig

* = Significant at 90%, 95% Confidence level, or Not Significant, and Exp(b) = Odd Ratio.

The gender variable (p-value = 0.498) was not significantly associated with the injury severity of the DDA. However, some previous studies reported that females had a greater tendency to be involved in DD compared to males [43,45]. Stimpson et al. [85] reported that 65.7% of pedestrian victims due to DD were male gender.

All vehicle-type categorical variables were statistically significant to the injury severity of the DDA, except the LMV category. The DDA severity due to the HMV (OR = 0.207) was lesser than the injury severity of the DDA due to the two-wheeler (OR = 1).

Two categorical variables of time of accident variable (i.e., 12AM-6AM and 6PM-12AM) were statistically significant to the injury severity of DDA. The severity of the DDA that occurred during night time driving (6PM-12AM) was 3.341 times higher than that of the DDA that occurred during dawn time driving (12AM-6AM) (OR = 1). The feasible explanation is the lack of visibility on road surfaces due to bright headlights from opposite vehicles and LED bulb-installed billboards; and driver fatigue and sleepiness. Our results are similar to previous research [45]. It was reported that most of the DDAs occurred during dawn and night time driving [45]. However, Stimpson et al. [85] reported that the number of motorist victims in the DDAs was high during morning and afternoon driving times (6AM-6PM).

In this study, only one category describing the nature of accident variable was statistically significant to the DDA severity: hit object. Head-on collision, side collision, rear-end collision, hit-pedestrians, and other collision-type were not significantly associated with the DDA severity. The severity of the DDA due to the hit-object collision type (OR = 0.588) was lesser than that due to the head-on collision type (OR = 1). However, some previous studies reported that rear-end collisions were strongly associated with a high probability of driver distraction [45,69].

## 4.4. BLR model-with-interaction results

The gender variable was found to be statistically insignificant to the DDA severity in the model-without-interaction. Therefore, it was not considered in the interaction analysis part. The results of significant categorical-interacted variables for two-way, three-way, and four-way interaction models are presented in Table 6. There were eleven significant combinations in the two-way-interaction model and one significant combination in the three-way as well as the four-way-interaction models.

### 4.4.1. Interaction between age and vehicle type.
There were two significant combinations (25–40 age group*HMV and above 40 age group*LMV) in the interaction between age and vehicle type variables (Table 6). The DDA severity for the middle-aged road users (25–40 age group) in the HMV category (OR = 0.273) was lesser than that for the under-aged road users (below 18 age group) in the two-wheelers category (OR = 1). Though lesser than that for the under-aged road users (below 18 age group) in two-wheelers, the DDA severity for the old-aged road users (Above 40 age group) in LMV category (OR = 0.608) was more than that for the middle-aged road users (25–40 age group) in HMV category (OR = 0.273).

### 4.4.2. Interaction between age and time of accident.
A similar pattern was observed in the two significant combinations (25−40 age group*12PM-6PM and above 40 age group*6AM-12PM) of the interaction analysis (between age and time of accident variables). The DDA severity for the old-aged road users (Above 40 age group) during the morning time (6AM-12PM) (OR = 0.290) was lesser than that for the under-aged road users (below 18 age group) during dawn time (12AM-6AM) (OR = 1). Though lesser than that for the under-aged road users (below 18 age group) during

**Table 6. Results of significant categorical-interacted variables.**

| Interacted Variables | b | S.E. | Wald | p-value | Sig* | Exp(b) |
|---|---|---|---|---|---|---|
| **Two-way-interaction** | | | | | | |
| *Age*Vehicle Type* | | | | | | |
| A(2) by VT(2) | −1.297 | 0.534 | 5.903 | 0.015 | 95% | 0.273 |
| A(3) by VT(1) | 0.498 | 0.296 | 2.837 | 0.092 | 90% | 0.608 |
| *Age*Time of accident* | | | | | | |
| A(2) by T(2) | −0.486 | 0.289 | 2.83 | 0.093 | 90% | 0.615 |
| A(3) by T(1) | −1.236 | 0.648 | 0.639 | 0.056 | 90% | 0.290 |
| *Age*Nature of accident* | | | | | | |
| A(1) by N(1) | −0.782 | 0.458 | 2.909 | 0.088 | 90% | 0.458 |
| A(1) by N(4) | 0.864 | 0.468 | 3.405 | 0.065 | 90% | 2.372 |
| A(2) by N(3) | −0.573 | 0.32 | 3.212 | 0.073 | 90% | 0.564 |
| A(2) by N(4) | 0.96 | 0.423 | 5.157 | 0.023 | 95% | 0.383 |
| A(3) by N(3) | −1.238 | 0.753 | 2.704 | 0.100 | 90% | 0.29 |
| *Time of accident*Vehicle Type* | | | | | | |
| T(2) by VT(2) | −2.045 | 1.025 | 3.979 | 0.046 | 95% | 0.129 |
| *Nature of accident*Time of accident* | | | | | | |
| N(3) by T(3) | −0.797 | 0.426 | 3.505 | 0.061 | 90% | 0.451 |
| **Three-way-interaction** | | | | | | |
| *Age*Time of accident*Vehicle Type* | | | | | | |
| A(2) by T(2) by VT(1) | −0.53 | 0.31 | 2.917 | 0.088 | 90% | 0.588 |
| **Four-way-interaction** | | | | | | |
| *Age*Nature of accident*Time of accident*Vehicle Type* | | | | | | |
| A(2)by N(1) by T(2) by VT(1) | 0.865 | 0.425 | 4.14 | 0.042 | 95% | 2.376 |

dawn time, The DDA severity for the middle-aged road users (25−40 age group) during afternoon time (12PM-6PM) (OR = 0.615) was more than that for the old-aged road users (above 40 age group) during the morning time.

**4.4.3. Interaction between age and nature of accident.** There were five significant combinations (18–24 age group*side collision, 18–24 age group*Hit pedestrian, 25–40 age group*hit object, 25–40 age group*Hit pedestrian, and above 40 age group*hit object) in the interaction between age and nature of accident variables (Table 6). It was observed that 18–24 age group* Hit pedestrian interaction (OR = 2.372) showed the strongest association with the injury severity of the DDA. This was followed by 25–40 age group*hit object (OR =0.564), 18–24 age group*side collision (OR = 0.458), 25–40 age group*Hit pedestrian (0.383), and above 40 age group*hit object (OR = 0.290) interaction. The DDA severities of the 25–40 age group*hit object, 18–24 age group*side collision, 25–40 age group*Hit pedestrian, and above 40 age group*hit object combinations were lower than that of the below 18 age group*head on collision interaction.

**4.4.4. Interaction between time of accident and vehicle type.** There was only one combination (12PM-6PM*HMV) that was observed to be significantly associated with the injury severity of the DDA. Results indicated that the injury severity of the DDA that occurred during afternoon time and HMV involved (12PM-6PM*HMV) (OR = 0.129) was lower than that of the DDA that occurred during dawn time and two-wheeler involved.

**4.4.5. Interaction between nature of accident and time of accident.** There was one significant combination (hit object*6PM-12AM) in the interaction between the nature of accident and time of accident variables (Table 9). The injury severity of the hit object-type accident due to the DD that occurred during night time (OR = 0.451) was lower than that of the head-on type accident due to the DD that occurred during dawn time.

**4.4.6. Interaction among age, time of accident, and vehicle type.** There was one significant combination in the Three-way interaction model, i.e., 25–40 age group*12PM-6PM*LMV (Table 6). The results showed that the injury severity of the DDA for the 25–40-years age group road users, where LMV was involved during the afternoon time (OR = 0.588), was lower than that of the DDA for the below 18-years age group road users, where two-wheeler was involved during dawn time (OR = 1).

**4.4.7. Interaction among age, nature of accident, time of accident, and vehicle type.** Similarly, there was also one significant combination in the Four-way interaction model, i.e., 25–40 age group*side collision*12PM-6PM*LMV (Table 6). The results revealed that the injury severity of the DDA for the 25–40-years age group road users, where the LMV type vehicle was involved in a side collision during afternoon time, was 2.376 times higher than that of the DDA for the below 18-years age group road users, where the two-wheeler was involved in the head-on collision during dawn time (OR = 1).

## 4.5. Machine Learning (ML) models results

The selection of the best ML model for the prediction of the injury severity of the DDAs was done based on the optimum combination of the performance metrics (accuracy, F1 score, and AUC). All six supervised ML algorithms were trained in two sets using the Dataiku platform. In the first set, all the algorithms were implemented using 5, 10, and 15-FCVs at a TR of 0.70. In the second set, all algorithms were trained at TRs of 0.70, 0.80, and 0.90 with 10-FCV. The selection of the best ML model based on the optimum combination of the performance metrics for the above-mentioned two sets is discussed in the following subsections.

**4.5.1. The best ML model for the first set.** Table 7 depicts the values of the performance metrics (accuracy, F1 score, and AUC) for 5-FCV at a TR of 0.70. The RF model showed the highest accuracy value (41.5%) among all the trained models in the case of 5-FCV. The SLP model ranked in the second position with an accuracy value of 40.8%. The KNN model was found to have the lowest accuracy value (26.8%). In terms of F1 score value, the RF model achieved the highest value (35.1%) among all six ML models. This was followed by the SLP and SVM models with an F1 score value of 35.0%. The KNN model presented the worst predictive performance based on the F1 score value. Further, the results of Table 10 also reveal that the XGBoost model showed the best performance in terms of the AUC value. This model obtained the highest AUC value (0.623) among all the trained models. The RF model was in the

**Table 7. Values of performance metrics for 5-FCV.**

| Sl. | ML algorithm | Accuracy | F1 score | AUC |
|---|---|---|---|---|
| 1 | RF | **0.415** | **0.351** | 0.586 |
| 2 | XGBoost | 0.309 | 0.349 | **0.623** |
| 3 | DT | 0.268 | 0.331 | 0.530 |
| 4 | KNN | 0.408 | 0.314 | 0.528 |
| 5 | SLP | 0.313 | 0.350 | 0.558 |
| 6 | SVM | 0.272 | 0.350 | 0.570 |

second rank with an AUC value of 0.586. The KNN model, with an AUC value (0.528), showed the worst performance in the prediction.

Table 8 depicts the values of the performance metrics (accuracy, F1 score, and AUC) for 10-FCV at a TR of 0.70. The results of Table 8 reveal that the XGBoost model presented a better performance than the other five trained models based on the accuracy value (57.0%) in the case of 10-FCV. This was followed by the RF and KNN models with accuracy values of 41.5% and 40.8%, respectively. The DT model stood in the last rank with an accuracy value (26.8%). In a previous study conducted by Ma et al. [52], it was reported that the SVM model achieved the lowest accuracy value of 86.27% when TR was 0.70. The F1 score metric values from Table 8 illustrate that the XGBoost model had the maximum F1 score value (39.4%), followed by the RF model (35.1%) and the SLP model (35.0%). The KNN model had the lowest F1 score value (31.4%) among all the models. Further, the results of Table 8 also reveal that the XGBoost model showed the best performance in terms of the AUC value (0.635). This was followed by the RF model (0.586), SVM (0.570), SLP (0.558), and DT (0.530). Similar to the case of the 5-FCV, the KNN model exhibited the lowest AUC value (0.528) in the case of the 10-FCV.

Table 9 depicts the values of the performance metrics (accuracy, F1 score, and AUC) for 15-FCV at a TR of 0.70. It was observed that the XGBoost (60.0% accuracy) model was the most accurate predictive model in the case of the 15-FCV. Similar to the case of the 10-FCV, the RF model (with an accuracy value of 41.5%) occupied the second position in the race in the case of the 15-FCV. This was followed by the KNN model (40.8%), the SLP model (31.3%), and the DT model

**Table 8. Values of performance metrics for 10-FCV.**

| Sl. | ML algorithm | Accuracy | F1 score | AUC |
|---|---|---|---|---|
| 1 | RF | 0.415 | 0.351 | 0.586 |
| 2 | XGBoost | **0.570** | **0.394** | **0.635** |
| 3 | DT | 0.268 | 0.331 | 0.530 |
| 4 | KNN | 0.408 | 0.314 | 0.528 |
| 5 | SLP | 0.313 | 0.350 | 0.558 |
| 6 | SVM | 0.230 | 0.338 | 0.570 |

**Table 9. Values of performance metrics for 15-FCV.**

| Sl. | ML algorithm | Accuracy | F1 score | AUC |
|---|---|---|---|---|
| 1 | RF | 0.415 | 0.351 | 0.586 |
| 2 | XGBoost | **0.600** | **0.376** | **0.627** |
| 3 | DT | 0.268 | 0.331 | 0.530 |
| 4 | KNN | 0.408 | 0.314 | 0.528 |
| 5 | SLP | 0.313 | 0.350 | 0.558 |
| 6 | SVM | 0.230 | 0.338 | 0.570 |

(26.8%). The SVM model had the lowest accuracy (23.0%) compared to the other trained models. In the case of the F1 score metric, the XGBoost model (37.6%) model achieved the best performance for the prediction of the DDA severity for 15-FCV. The RF model (35.1%) occupied the second position, followed by the SLP model with an F1 score value of 35.0%. The KNN model was found to have the lowest F1 score value (31.4%). As seen in Table 9, the XGBoost model had the highest AUC value (0.627) similar to that of the 5- and 10-FCVs. This was followed by the RF model with an AUC value of 0.586. The KNN model exhibited the lowest AUC value (0.528).

The overall summary of the best ML model performance for the first set is presented in Table 10. It is observed that the RF model can be selected as the best ML model for the prediction of the DDA severity in the case of the 5-FCV. However, for the 10- and 15-FCV, it is seen that the XGBoost model can be considered the best ML model. Overall, it can be concluded that with the first set condition, the XGBoost model can be applied in Imphal city for the most accurate prediction of the DDA severity.

**4.5.2. The best ML model for the second Set.** In the second set, three hyperparameter conditions were applied: 10-fold with 0.70 TR, 10-fold with 0.80 TR, and 10-fold with 0.90TR. Since the results of the first hyperparameter condition (10-fold with 0.70 TR) are already presented in Table 8, analysis and discussion of this condition will be same and are not mentioned in this section. However, the name of the best ML model for 10-fold with TR 0.70 is listed in Table 13.

Table 11 depicts the values of the performance metrics for 10-FCV at a TR of 0.80. It was observed that the SLP (60.0% accuracy) model was found to be the most accurate predictive model for TR 0.80. However, different results have been observed in previous studies [33,52,55,61]. It was reported that the Genetic Search algorithm presented the best performance with 67.8% prediction accuracy for driver distraction [55]. Bachtiar et al. [61] reported that the RF model presented a better performance than all other models for the prediction of driver distraction with an accuracy of 98.85%. This observation was congruent with that of Misra et al. [33]. The DT model ranked in the second position with an accuracy value of 57.3%. The XGBoost model stood in the last position with an accuracy value of 21.6%. In the F1 score metric part, the SLP model showed a better performance among six ML models with an F1 score value of 35.1%. The SVM model ranked in the second position with an F1 score of 34.9%. The KNN model presented the lowest performance results based on the F1 score value for TR 0.80. In a previous study [32], it was observed that the XGBoost model obtained the best performance with an F1 score value of 94.0%. Liao et al. [49] reported the SVM model had the highest predictive performance with an F1 score value of 95.8%. Further, the comparison results also show that the SLP model

**Table 10. Summary of the best model performance for the first set.**

| TR | k-fold | Best model | Performance metrics | | |
|----|--------|------------|---------------------|---|---|
| | | | Accuracy | F1 score | AUC |
| 0.7 | 5 | RF | **0.415** | **0.351** | 0.586 |
| | 10 | XGBoost | **0.570** | **0.394** | **0.635** |
| | 15 | XGBoost | **0.600** | **0.376** | **0.627** |

**Table 11. Values of performance metrics for TR 0.80.**

| Sl. | ML algorithm | Accuracy | F1 score | AUC |
|-----|--------------|----------|----------|-----|
| 1 | RF | 0.470 | 0.338 | 0.577 |
| 2 | XGBoost | 0.216 | 0.332 | 0.574 |
| 3 | DT | 0.573 | 0.347 | 0.552 |
| 4 | KNN | 0.411 | 0.297 | 0.515 |
| 5 | SLP | **0.600** | **0.351** | **0.611** |
| 6 | SVM | 0.314 | 0.349 | 0.547 |

had a maximum AUC value (0.611) indicating a better performance compared to all other trained models. The RF model stood in the second rank with an AUC value of 0.577. This was followed by the XGBoost model (0.574). The KNN model showed the worst performance with the lowest AUC value (0.517). However, in a simulator study conducted by Koohestani et al. [22], it was reported that the RF model had a maximum AUC value (0.92), outperforming the other ML models.

The XGBoost model had the highest accuracy of 77.4% for TR 0.90, as shown in Table 12. This was followed by the SLP model with an accuracy value of 67.7%. The SVM model exhibited the lowest performance based on the accuracy value (36.6%). The SLP model presented the best performance with a maximum F1 score value of 42.3%. This was followed by the DT model with an F1 score value of 41.9%. The KNN model was in the last position of the race with an F1 score value of 26.7%. The DT model had a maximum AUC value (0.656) among all trained models. The SLP model attained the second position with an AUC value of 0.748. This was followed by the RF model (0.624). The KNN model exhibited the lowest performance results in terms of the AUC value. Mcdonald et al. [53] reported the RF model as the best-performing model in terms of accuracy and AUC metrics for predicting driver distraction.

The overall summary of the best ML model performance for the second set is presented in Table 13. It is observed that the XGBoost model was the best ML model for the prediction of the DDA severity for TR of 0.70. However, for train 0.80 and 0.90, the SLP model can be considered the best ML model for the prediction of injury severity of the DDA in Imphal city. Lastly, according to the second set condition, the SLP model can be applied in Imphal city for the most accurate prediction of the DDA severity.

## 5. Summary and conclusions

Road traffic accidents (RTAs) are one of the most significant causes of serious injuries and deaths to road users worldwide. The increase in the number of RTAs due to driver distraction is one of the concerns among transportation communities. The present study aims to examine the individual and interacted effects of the influential factors on the injury severity of DD accidents (DDAs) using a Binary Logistic Regression (BLR) method, and at the same, to select the best machine learning (ML) model for the prediction of the injury severity of the DDA using Dataiku platform. Police-reported DD data collected for Imphal city during 2011–2020 was used in the present study. This dataset consisted of 949 observations that included single- and two-vehicle accident data only. In the BLR method, two sets of models (Model-without-interaction and Model-with-interaction: two-way and three-way interaction) were developed using SPSS software to examine the effect of contributing factors, along

**Table 12. Values of performance metrics for TR 0.90.**

| Sl. | ML algorithm | Accuracy | F1 score | AUC |
|---|---|---|---|---|
| 1 | RF | 0.419 | 0.386 | 0.624 |
| 2 | XGBoost | **0.774** | 0.400 | 0.573 |
| 3 | DT | 0.613 | 0.419 | **0.656** |
| 4 | KNN | 0.409 | 0.267 | 0.430 |
| 5 | SLP | 0.677 | **0.423** | 0.648 |
| 6 | SVM | 0.366 | 0.352 | 0.577 |

**Table 13. Summary of the best model performance for the second set.**

| K-fold | TR | Best model | Performance metrics | | |
|---|---|---|---|---|---|
| | | | Accuracy | F1 score | AUC |
| 10 | 0.70 | XGBoost | 0.570 | 0.394 | 0.635 |
| | 0.80 | SLP | **0.600** | **0.351** | **0.611** |
| | 0.90 | SLP | 0.677 | **0.423** | 0.648 |

with their interaction effect, on the injury severity of the DDA. The selection of the best ML model for the prediction of the injury severity of the DDAs was done based on the best combination of accuracy, F1 score, and AUC metrics. All six supervised ML algorithms (RF, DT, XGBoost, KNN, SVM, and SLP) were trained in two sets using the Dataiku platform. In the first set, all the algorithms were implemented using 5, 10, and 15-FCVs at a TR of 0.70. In the second set, all algorithms were trained at TRs of 0.70, 0.80, and 0.90 with 10-FCV. The important conclusions drawn from the study are summarized below.

## 5.1. From the BLR model results

### 5.1.1. BLR model-without-interaction.

- Eight categorical variables (below 18 age group, 18−24 age group, 25−40 age group, above 40 years age group, Two-wheeler, HMV, 6PM-12AM, and hit-object collision) were statistically significant to the injury severity of the DDA at a 95% confidence level, and one categorical variable (12AM-6AM) was significant at a 90% confidence level.

- The injury severity of the DDA for the below 18-year age group road users was more than that of the 18–24, 25–40, and Above 40-years age group road users (OR =0.483, 0.490, and 0.304, respectively).

- The gender variable was not statistically significant to the injury severity of the DDA.

- The DDA severity due to the HMVs (OR = 0.207) was lesser than the injury severity of the DDAs due to the two-wheelers (OR = 1).

- The severity of the DDA that occurred during night time driving (6PM-12AM) was 3.341 times higher than that of the DDA that occurred during dawn time driving (12AM-6AM) (OR = 1).

- The severity of the DDA due to the hit-object collision type (OR = 0.588) was lesser than that due to the head-on collision type (OR = 1).

### 5.1.2. BLR model-with-interaction.

- There were eleven significant combinations in the Two-way-interaction model, and one significant combination in the Three-way as well as in the Four-way-interaction models.

- The DDA severity for the under-aged road users (below 18 age group), where the two-wheeler was involved, was higher than that for the middle-aged road users (25–40 age group) with HMV (OR = 0.273) and that for the old-aged road users (Above 40 age group) with LMV vehicle type (OR = 0.608).

- The DDA severity for the under-aged road users (below 18 age group) during dawn time (OR = 1) was higher than that for the middle-aged road users (25–40 age group) during afternoon time (OR = 0.615) and that for the old-aged drivers (Above 40 age group) during morning time (OR = 0.290).

- The injury severity of the DDA for the 18–24-years age group road users who got into hit-pedestrian accidents was 2.372 times higher than that for the below 18-years age group road users who were involved in head-on collision-type accidents. The injury severity for the above 40-years age group road users who were involved in hit-object type accidents (OR = 0.290) was the lowest when compared to the below 18-years age group road users who were involved in head-on collision-type accidents.

- The injury severity of the DDA that occurred during afternoon time and HMV involved (12PM-6PM*HMV) (OR = 0.129) was lower than that of the DDA that occurred during dawn time and two-wheeler involved.

- The injury severity of the hit object-type accident due to the DD that occurred during night time (OR = 0.451) was lower than that of the head-on type accident due to the DD that occurred during dawn time.

- In the three-way interaction results, it was observed that the injury severity of the DDA for 25–40-years age group road users, where LMV type vehicle was involved during the afternoon time (OR = 0.588), was lower than that of the DDA for the below 18-years age group road users, where two-wheeler was involved during dawn time (OR = 1).

- In the four-way interaction results, the injury severity of the DDA for the 25–40-years age group road users, where LMV type vehicle was involved in a side collision during afternoon time, was 2.376 times higher than that of the DDA for the below 18-years age group road users, where two-wheeler was involved in the head-on collision during dawn time (OR = 1).

### 5.2. From the ML models results

#### 5.2.1. From the first set.

- The RF model was found to be the best ML model for the prediction of the DDA severity in the case of the 5-FCV.

- The XGBoost model could be considered the best ML model for the prediction of injury severity of the DDA in Imphal city for the 10- and 15-FCV.

- Overall, the XGBoost model can be used for the most accurate prediction of the DDA severity for the city.

#### 5.2.2. From the second set.

- The XGBoost model was found to be the best ML model for the prediction of the DDA severity for a TR of 0.70.

- For train 0.80 and 0.90, the SLP model could be applied for the most accurate prediction of the DDA severity for the city.

- Overall, the SLP model can be considered the best ML model for the prediction of injury severity of the DDA for the city.

### 5.3. Limitations and future research

This study is based on the police-reported data; so, there is a chance of under-reporting of distracted near-accident type data. However, the most commonly used datasets have been obtained from either naturalistic driving or simulator studies in the field of distraction research. The data used in this study includes fatal and non-fatal accidents only. It does not include detailed injury levels: property damage only, visible injury, and serious injury. The treatment of class imbalance during the model training process is another limitation of the current study. While techniques such as oversampling, under-sampling, or class-weight adjustments could have been applied to mitigate the impact of class imbalance, these methods were not implemented in the present study. As a result, the performance of the models may be biased towards the majority class, potentially limiting their ability to accurately predict instances from the minority class. In future studies, class imbalance mitigation strategies could be implemented to achieve better robustness of the models and more consistent prediction results. Additionally, future work should explore more advanced approaches like deep learning on more comprehensive accident datasets from other states of the Northeast region, with the injury severity of individual road users as the target variable.

## Supporting information

**S1 File. Data used in the study.**
(XLSX)

## Acknowledgments

The authors like to extend their gratitude towards the Superintendents of Police, Traffic Cell, Department of police for Imphal city, India for providing police accident report data of the city.

## Author contributions

**Conceptualization:** Neero Gumsar Sorum.

**Data curation:** Martina Gumsar Sorum.

**Formal analysis:** Martina Gumsar Sorum.

**Investigation:** Martina Gumsar Sorum.

**Methodology:** Neero Gumsar Sorum, Martina Gumsar Sorum.

**Project administration:** Neero Gumsar Sorum.

**Resources:** Neero Gumsar Sorum.

**Software:** Martina Gumsar Sorum.

**Supervision:** Neero Gumsar Sorum.

**Validation:** Neero Gumsar Sorum.

**Visualization:** Martina Gumsar Sorum.

**Writing – original draft:** Martina Gumsar Sorum.

**Writing – review & editing:** Neero Gumsar Sorum.

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
