## [Decision Letter · Decision Letter 0]

Mar 21 2025

PONE-D-24-45051Modeling of Injury Severity of Distracted Driving Accident Using Statistical and Machine Learning ModelsPLOS ONE

Dear Dr. sorum,

Thank you for submitting your manuscript to PLOS ONE. After careful consideration, we feel that it has merit but does not fully meet PLOS ONE’s publication criteria as it currently stands. Therefore, we invite you to submit a revised version of the manuscript that addresses the points raised during the review process.

Please try to revise your manuscript and respond to all the reviewers' comments.

We look forward to receiving your revised manuscript.

Kind regards,

Quan Yuan, Ph.D.

Academic Editor

PLOS ONE

Journal Requirements:

2. a. For studies reporting research involving human participants, PLOS ONE requires authors to confirm that this specific study was reviewed and approved by an institutional review board (ethics committee) before the study began. Please provide the specific name of the ethics committee/IRB that approved your study, or explain why you did not seek approval in this case.

b. Please provide additional details regarding participant consent. In the ethics statement in the Methods and online submission information, please ensure that you have specified (1) whether consent was informed and (2) what type you obtained (for instance, written or verbal, and if verbal, how it was documented and witnessed). If your study included minors, state whether you obtained consent from parents or guardians. If the need for consent was waived by the ethics committee, please include this information.

Reviewers' comments:

Reviewer's Responses to Questions

**Comments to the Author**

1. Is the manuscript technically sound, and do the data support the conclusions?

Reviewer #1: Yes

Reviewer #2: Yes

2. Has the statistical analysis been performed appropriately and rigorously? 

Reviewer #1: Yes

Reviewer #2: Yes

3. Have the authors made all data underlying the findings in their manuscript fully available?

Reviewer #1: No

Reviewer #2: Yes

4. Is the manuscript presented in an intelligible fashion and written in standard English?

Reviewer #1: Yes

Reviewer #2: Yes

5. Review Comments to the Author

Reviewer #1: This paper examines the individual and interacted effects of the influential factors on the injury severity of the DDAs using the Binary Logistic Regression (BLR) method, and to select the best machine learning (ML) model in predicting the injury severity of the DDA. The study is based on the 10-year accident data from India.

Overall, the paper is well-organized and easy to follow. However, I have the following concerns.

Major:

1. The dataset consists of approximately 78.6% non-fatal accidents and 21.4% fatal accidents, suggesting that a classification model could easily achieve an accuracy of at least 0.78 by simply predicting non-fatal accidents. However, the model in the paper yields an accuracy ranging from 0.4 to 0.77. Could the authors provide a more detailed explanation of these results, particularly regarding why the model does not perform as expected given the huge class imbalance?

2. The age groups are categorized as follows: below 18, 18-24, 25-40, and above 40. The second threshold is set at 24 (years old), which seems somewhat arbitrary. Could the authors clarify why this specific threshold was chosen? Furthermore, the third age group (25-40) contains the largest number of samples, which may lead to the dominance of this group in the analysis. This could skew the results. Could the authors consider either adjusting the groupings or discussing this potential bias?

Minor:

1. The study concludes that the "gender variable was not statistically significant to the injury severity of the DDA," which contrasts with findings from other studies. Could the authors provide an explanation for this discrepancy? It would be helpful to understand why gender does not appear to be a significant factor in this case.

2. On Page 19, Line 525, the authors apply three different hyperparameter conditions: 10-fold cross-validation with training sets of 70%, 80%, and 90%. Could the authors clarify why 10-fold cross-validation was chosen here? Based on the earlier results, 15-fold cross-validation seemed to yield the highest accuracy.

Reviewer #2: 1.How did the author confirm that the accident cases used were caused by distracted driving? If this problem cannot be solved, the core of this research will be seriously challenged.

2. Although 20 categorical variables have been identified, I think there is no deep connection between the involved variables and distracted driving. Is it possible to add factors such as using a mobile phone, operating the car's display screen, feeling sleepy, chatting, etc.?

3. The machine learning model only uses the default settings of the Dataiku platform and does not fully explore the optimal parameter combinations.

4. The performance of the model in dealing with the imbalance between the number of fatal and non-fatal accidents has not been deeply explored.

5. The time trend change of DDA data during 2011- 2020 has not been analyzed. Time series analysis can be added to study the variation law of accident severity over time and the dynamic influence of influencing factors.

6. Can the influence of environmental factors be considered?

6. PLOS authors have the option to publish the peer review history of their article (what does this mean? ). If published, this will include your full peer review and any attached files.

**Do you want your identity to be public for this peer review?** For information about this choice, including consent withdrawal, please see our Privacy Policy .

Reviewer #1: No

Reviewer #2: No

---

## [Author Response · Author response to Decision Letter 1]

11 Feb 2025

Reply to Academic Editor and Reviewer’s Comments

Reply to the reviewer’s comments on the manuscript entitled ‘Modeling of Injury Severity of Distracted Driving Accident Using Statistical and Machine Learning Models", by Neero Gumsar Sorum and Martina Gumsar Sorum. Thanks to the Academic Editor, “PLOS ONE)” for allowing us to improve the current version of the manuscript and resubmission after revision. The authors would also like to sincerely thank all the anonymous reviewers for their encouraging and constructive comments to improve the quality of the manuscript.

Comments from the Academic Editor and Reviewers:

Academic Editors Comments to Authors

Reply:

The revised manuscript is prepared as per the PLOS ONE's style requirements, including those for file naming.

2) a. For studies reporting research involving human participants, PLOS ONE requires authors to confirm that this specific study was reviewed and approved by an institutional review board (ethics committee) before the study began. Please provide the specific name of the ethics committee/IRB that approved your study, or explain why you did not seek approval in this case.

b. Please provide additional details regarding participant consent. In the ethics statement in the Methods and online submission information, please ensure that you have specified (1) whether consent was informed and (2) what type you obtained (for instance, written or verbal, and if verbal, how it was documented and witnessed). If your study included minors, state whether you obtained consent from parents or guardians. If the need for consent was waived by the ethics committee, please include this information.

Reply: Not Applicable.

3) We note that you have indicated that there are restrictions to data sharing for this study. PLOS only allows data to be available upon request if there are legal or ethical restrictions on sharing data publicly. For more information on unacceptable data access restrictions, please see http://journals.plos.org/plosone/s/data-availability#loc-unacceptable-data-access-restrictions.

Reply:

Actually, we couldn't understand this statement properly at the time of article submission. Now understood, and the dataset used for the present study is uploaded as a supporting information file.

Reviewers Comment to Authors:

Reviewer 1

## This paper examines the individual and interacted effects of the influential factors on the injury severity of the DDAs using the Binary Logistic Regression (BLR) method, and to select the best machine learning (ML) model in predicting the injury severity of the DDA. The study is based on the 10-year accident data from India. Overall, the paper is well-organized and easy to follow. However, I have the following concerns.

Thanks for the encouraging comments. The concerns in the manuscript are incorporated point by point and given below.

Major

1) The dataset consists of approximately 78.6% non-fatal accidents and 21.4% fatal accidents, suggesting that a classification model could easily achieve an accuracy of at least 0.78 by simply predicting non-fatal accidents. However, the model in the paper yields an accuracy ranging from 0.4 to 0.77. Could the authors provide a more detailed explanation of these results, particularly regarding why the model does not perform as expected given the huge class imbalance?

Reply:

In the context of imbalanced datasets, accuracy is often not the best measure of model performance, as it can be misleading. A model might predict the majority class most of the time and still achieve a high accuracy while performing poorly on the minority class (fatal accidents). Therefore, F1-Score and AUC metrics were also used for the evaluation of model performance in the present study. These metrics provide a more detailed understanding of how well the model is distinguishing between fatal and non-fatal accidents, especially in the case of imbalanced data. The possible reason for the actual performance of the ML model (accuracy =0.4 to 0.77) in the present study might be due to overfitting, underfitting, or a lack of variables that effectively capture the key differences between the non-fatal and fatal classes.

2). The age groups are categorized as follows: below 18, 18-24, 25-40, and above 40. The second threshold is set at 24 (years old), which seems somewhat arbitrary. Could the authors clarify why this specific threshold was chosen? Furthermore, the third age group (25-40) contains the largest number of samples, which may lead to the dominance of this group in the analysis. This could skew the results. Could the authors consider either adjusting the groupings or discussing this potential bias?

Reply:

It was planned to categorize the age group variable into teen age group (below 18), young age group (18-30), middle-aged group (31-45), and old age group (above 45) for the study. however, after analyzing, it was difficult to make group of middle-aged group data because young age group and old age group had higher numbers in the original data received (the dataset used in the present study is only for distracted driving: one cause of accident). Therefore, the age grouping was re-arranged into below 18, 18-24, 25-40, and above 40 so that each group could make a more or less balanced representation.

Minor:

1) The study concludes that the "gender variable was not statistically significant to the injury severity of the DDA," which contrasts with findings from other studies. Could the authors provide an explanation for this discrepancy? It would be helpful to understand why gender does not appear to be a significant factor in this case.

Reply:

The results of the present study indicated gender variable was not statistically significant to the injury severity of the DDA (p-value > 0.05). The statistical insignificance indicates that gender does not have a measurable impact on the injury severity of DDA. The possible explanation may be that other variables (e.g., age, vehicle type, nature of accident, and time of accident) played a much larger role in injury severity, and their effects might overshadow the potential impact of gender. When multiple factors are accounted for in the model, gender may not show any significant relationship. Another possible reason might be due to the imbalanced dataset (e.g., predominantly one gender). It may lead to an inability to detect any significant differences.

2) On Page 19, Line 525, the authors apply three different hyperparameter conditions: 10-fold cross-validation with training sets of 70%, 80%, and 90%. Could the authors clarify why 10-fold cross-validation was chosen here? Based on the earlier results, 15-fold cross-validation seemed to yield the highest accuracy.

Reply:

The main objective of hyperparameter tuning (changing k-fold cross-validation and train ratio values) was to study the variation in model performance, accordingly, the best ML model will change. So, in the first set, the ML algorithms were implemented by changing k-fold cross-validation values (5-, 10-, and 15-FCV) at a train ratio value (TR = 0.7), and in the second set, the algorithms were trained at three different TR values (0.7, 0.8, and 0.9) at a 10-FCV. Further, the two sets used in the present study were completely independent. In other words, it means that the first set can be carried out after the second set (in that case, the first set will become the second set) and vice-versa.

Reviewer 2

1) How did the author confirm that the accident cases used were caused by distracted driving? If this problem cannot be solved, the core of this research will be seriously challenged

Reply:

According to the police-reported accident data (used in the present study), those accidents which occurred due to mobile use, talking with passengers, or looking away from the roadway, were commonly designated as distracted driving accidents (DDAs). According to the police officials, the officer in charge asked the passengers (if involved in the accident) or the drivers (who survived the accidents) about the cause of the accident (on the spot or during medical treatment). If it was due to mobile use while driving, chatting/talking, or looking away from the roadway, then that particular accident was designated as a distracted driving accident (DDA). For more clarity, the following sentences are incorporated in the revised manuscript and highlighted:

“In the present study, the DDAs were defined as those accidents which occurred due to driver engagement in secondary tasks (mobile use, chatting and talking, or looking away from the roadway) other than driving.” (line numbers: 303-305, page number: 14)

“The dataset contained two DDA categories: Non-fatal (not resulting in death within one month of the accident) and fatal (resulting in death within one month of the accident).” (line numbers: 308-310, page number: 14 )

2) Although 20 categorical variables have been identified, I think there is no deep connection between the involved variables and distracted driving. Is it possible to add factors such as using a mobile phone, operating the car's display screen, feeling sleepy, chatting, etc.?

Reply:

The categorical variables identified in the present study were not the causes of distracted driving but were the factors that might contribute to the injury severity of the DDAs. Therefore, the main aims of the present study were (i) to examine the individual and interacted effects of these identified factors on the injury severity of the DDA using a logistic regression method, and (ii) to select the best ML model in predicting the injury severity of the DDA among trained six ML models.

3) The machine learning model only uses the default settings of the Dataiku platform and does not fully explore the optimal parameter combinations.

Reply:

By “The default settings of the Dataiku platform” statement, the authors meant that the ML algorithms available in the Dataiku platform were used, and no new algorithms were developed by their own in the platform. Regarding optimal parameter combinations, two sets of hyperparameter tuning were employed in the present study (mentioned in line numbers:…. And page number: …….):

In Set 1, all six ML algorithms were implemented using 5, 10, and 15-fold cross-validation in a Train Ratio value of 0.70.

In Set 2, the ML algorithms were trained at Train Ratio values of 0.70, 0.80, and 0.90 with 10-fold cross-validation.

4) The performance of the model in dealing with the imbalance between the number of fatal and non-fatal accidents has not been deeply explored.

Reply:

To deal with the imbalance between the number of fatal and non-fatal accidents, the present study employed a stratified k-fold cross-validation technique (instead of under-sampling and oversampling techniques) because this technique addresses this issue by ensuring that each fold contains approximately the same proportion of each class as the entire dataset. This is particularly important in imbalanced datasets where one class (e.g., non-fatal accidents) may be significantly more prevalent than the other (e.g., fatal accidents). By maintaining the class distribution in each fold, the stratified k-fold helps the model learn better representations for both classes and avoids bias toward the majority class.

This is described in section 3.6 of the revised manuscript (page 17).

5) The time trend change of DDA data during 2011- 2020 has not been analyzed. Time series analysis can be added to study the variation law of accident severity over time and the dynamic influence of influencing factors.

Reply:

The study of temporal stability/instability of contributing factors to the injury severity of DDA was not included in the present study and the same will be taken up in future research by the authors.

6) Can the influence of environmental factors be considered?

Reply:

There was no information about environmental factors in the police-reported accident data used in the present study, otherwise, these factors would have been included.

---

## [Decision Letter · Decision Letter 1]

Mar 21 2025

PONE-D-24-45051R1Modeling of Injury Severity of Distracted Driving Accident Using Statistical and Machine Learning ModelsPLOS ONE

Dear Dr. sorum,

Thank you for submitting your manuscript to PLOS ONE. After careful consideration, we feel that it has merit but does not fully meet PLOS ONE’s publication criteria as it currently stands. Therefore, we invite you to submit a revised version of the manuscript that addresses the points raised during the review process.

**Please address the reviewer's new comments and revise the manuscript again.**

We look forward to receiving your revised manuscript.

Kind regards,

Quan Yuan, Ph.D.

Academic Editor

PLOS ONE

Reviewers' comments:

Reviewer's Responses to Questions

**Comments to the Author**

1. If the authors have adequately addressed your comments raised in a previous round of review and you feel that this manuscript is now acceptable for publication, you may indicate that here to bypass the “Comments to the Author” section, enter your conflict of interest statement in the “Confidential to Editor” section, and submit your "Accept" recommendation.

Reviewer #1: (No Response)

Reviewer #2: All comments have been addressed

2. Is the manuscript technically sound, and do the data support the conclusions?

Reviewer #1: Yes

Reviewer #2: Yes

3. Has the statistical analysis been performed appropriately and rigorously? 

Reviewer #1: Yes

Reviewer #2: Yes

4. Have the authors made all data underlying the findings in their manuscript fully available?

Reviewer #1: Yes

Reviewer #2: Yes

5. Is the manuscript presented in an intelligible fashion and written in standard English?

Reviewer #1: Yes

Reviewer #2: Yes

6. Review Comments to the Author

**Reviewer #1:**  While the authors claim to have handled the imbalance using a stratified k-fold cross-validation technique, this reflects a misunderstanding of the method. Stratified k-fold ensures proportional representation of classes in each fold but does not actively address imbalance during model training.

Additionally, the machine learning models exhibit poor performance, even when considering F1-score and AUC. The results suggest that the models are performing only slightly better than random, indicating that they fail to effectively distinguish between non-fatal and fatal cases. Addressing the class imbalance properly and improving model robustness are necessary to enhance predictive performance. Without these improvements, the value of the ML approach remains questionable

**Reviewer #2: ** The authors have addressed all the comments point-by-point in their revised manuscript. The revised article now provides a detailed explanation of the DDA (distracted driving accident) determination criteria and data sources, which enhances the credibility of the study. The methodology is rigorous, with reasonable solutions proposed for addressing class imbalance and hyperparameter tuning. The authors have also acknowledged the limitations of the data (e.g., the absence of environmental factors) and expressed their intention to conduct future research on temporal trends. Overall, I have no further suggestions. It is recommended that the revised manuscript be accepted.

7. PLOS authors have the option to publish the peer review history of their article (what does this mean? ). If published, this will include your full peer review and any attached files.

**Do you want your identity to be public for this peer review?** For information about this choice, including consent withdrawal, please see our Privacy Policy .

Reviewer #1: No

Reviewer #2: **Yes: ** wei ji

---

## [Author Response · Author response to Decision Letter 2]

10 Apr 2025

Comments from the Academic Editor and Reviewers:

Academic Editors Comments to Authors

## Dear Dr. Sorum,

Thank you for submitting your manuscript to PLOS ONE. After careful consideration, we feel that it has merit but does not fully meet PLOS ONE’s publication criteria as it currently stands. Therefore, we invite you to submit a revised version of the manuscript that addresses the points raised during the review process.

Reply:

The latest revised manuscript has been prepared by incorporating the concerns raised by the reviewers.

Reviewers Comment to Authors:

Reviewer 1

##

While the authors claim to have handled the imbalance using a stratified k-fold cross-validation technique, this reflects a misunderstanding of the method. Stratified k-fold ensures proportional representation of classes in each fold but does not actively address imbalance during model training.

Additionally, the machine learning models exhibit poor performance, even when considering F1-score and AUC. The results suggest that the models are performing only slightly better than random, indicating that they fail to effectively distinguish between non-fatal and fatal cases. Addressing the class imbalance properly and improving model robustness are necessary to enhance predictive performance. Without these improvements, the value of the ML approach remains questionable

Reply:

Thank you for the insightful feedback. While it is true that stratified k-fold cross-validation ensures proportional representation of classes in each fold, we agree that it does not address class imbalance during model training. The authors' claim may reflect a misunderstanding of the method’s limitations in handling imbalance in terms of model learning - To address this, we could consider employing techniques like oversampling, undersampling, or using class-weight adjustments during training, which can more directly mitigate the impact of imbalance on model performance. Since the issue of imbalance in the dataset was not handled in the present study, this limitation is mentioned/addressed in the limitation section of the latest revised manuscript (the same is highlighted and incorporated in section 5.3 of the latest revised manuscript).

Major

Reviewer 2

1) The authors have addressed all the comments point-by-point in their revised manuscript. The revised article now provides a detailed explanation of the DDA (distracted driving accident) determination criteria and data sources, which enhances the credibility of the study. The methodology is rigorous, with reasonable solutions proposed for addressing class imbalance and hyperparameter tuning. The authors have also acknowledged the limitations of the data (e.g., the absence of environmental factors) and expressed their intention to conduct future research on temporal trends. Overall, I have no further suggestions. It is recommended that the revised manuscript be accepted.

Reply:

Thank you for your encouraging comments.

---

## [Decision Letter · Decision Letter 2]

Modeling of Injury Severity of Distracted Driving Accident Using Statistical and Machine Learning Models

PONE-D-24-45051R2

Dear Dr. sorum,

We’re pleased to inform you that your manuscript has been judged scientifically suitable for publication and will be formally accepted for publication once it meets all outstanding technical requirements.

Kind regards,

Quan Yuan, Ph.D.

Academic Editor

PLOS ONE

Additional Editor Comments (optional):

Reviewers' comments:

Reviewer's Responses to Questions

**Comments to the Author**

1. If the authors have adequately addressed your comments raised in a previous round of review and you feel that this manuscript is now acceptable for publication, you may indicate that here to bypass the “Comments to the Author” section, enter your conflict of interest statement in the “Confidential to Editor” section, and submit your "Accept" recommendation.

Reviewer #1: All comments have been addressed

2. Is the manuscript technically sound, and do the data support the conclusions?

Reviewer #1: Yes

3. Has the statistical analysis been performed appropriately and rigorously? 

Reviewer #1: Yes

4. Have the authors made all data underlying the findings in their manuscript fully available?

Reviewer #1: Yes

5. Is the manuscript presented in an intelligible fashion and written in standard English?

Reviewer #1: Yes

6. Review Comments to the Author

Reviewer #1: The authors have responded to all comments point by point in the revised draft. The revised draft explicitly includes the handling of unbalanced datasets as a limitation, which is fair.

7. PLOS authors have the option to publish the peer review history of their article (what does this mean? ). If published, this will include your full peer review and any attached files.

**Do you want your identity to be public for this peer review?** For information about this choice, including consent withdrawal, please see our Privacy Policy .

Reviewer #1: No

---

## [Editor Report · Acceptance letter]

PONE-D-24-45051R2

PLOS ONE

Dear Dr. sorum,

I'm pleased to inform you that your manuscript has been deemed suitable for publication in PLOS ONE. Congratulations! Your manuscript is now being handed over to our production team.

Kind regards,

on behalf of

Dr. Quan Yuan

Academic Editor

PLOS ONE